# Using Satellite Data to Determine Empirical Relationships between Volcanic Ash Source Parameters

**Meelis J. Zidikheri * and Chris Lucas** 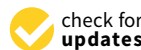

Australian Bureau of Meteorology, Melbourne, Victoria 3001, Australia; chris.lucas@bom.gov.au

* Correspondence: meelis.zidikheri@bom.gov.au

**Abstract:** Poor knowledge of dispersion model source parameters related to quantities such as the total fine ash mass emission rate, its effective spatial distribution, and particle size distribution makes the provision of quantitative forecasts of volcanic ash a difficult problem. To ameliorate this problem, we make use of satellite-retrieved mass load data from 14 eruption case studies to estimate fine ash mass emission rates and other source parameters by an inverse modelling procedure, which requires multidimensional sampling of several thousand trial simulations with different values of source parameters. We then estimate the dependence of these optimal source parameters on eruption height. We show that using these empirical relationships in a data assimilation procedure leads to substantial improvements to the forecasts of ash mass loads, with the use of empirical relationships between parameters and eruption height having the added advantage of computational efficiency because of dimensional reduction. In addition, the use of empirical relationships, which encode information in satellite retrievals from past case studies, implies that quantitative forecasts can still be issued even when satellite retrievals of mass load are not available in real time due to cloud cover or other reasons, making it especially useful for operations in the tropics where ice and water clouds are ubiquitous.

**Keywords:** volcanic ash; inverse modelling; parameter estimation; quantitative forecasts; ensemble modelling; dispersion modelling; source estimation

## 1. Introduction

Due to the hazards posed by volcanic ash on airborne aircraft, the monitoring and forecasting of volcanic ash is a key service for the aviation industry [1]. The Australian Bureau of Meteorology provides this service through the Darwin Volcanic Ash Advisory Centre (VAAC), one of nine such centers globally. It is responsible for issuing volcanic ash advisories in the volcanically active region covering Indonesia, Papua New Guinea, and the southern Philippines. The volcanic ash advisories issued by the VAACs comprise polygon coordinates that identify the regions of contaminated airspace. These advisories are qualitative in the sense that no estimates of the amount of ash in the contaminated region are provided. However, because of the potentially high economic impact of eruption events on airline operations, as for example during the high-impact Icelandic Eyjafjallajökull volcanic eruption event of 2010 [2], there has been an increasing demand for quantitative forecasts that provide information on the amount of ash in the contaminated regions. Quantitative forecasts will enable aircraft operators to better manage the risks of flying through regions of airspace with some likelihood of ash contamination.

Dispersion models, which are driven by gridded fields such as wind and rainfall obtained from meteorological Numerical Weather Prediction (NWP) models, are the primary tool for issuing volcanic ash forecasts. However, using these tools to provide reliable quantitative forecasts of airborne volcanic

ash is a challenging problem. A major source of uncertainty lies in the specification of the 'source term', which includes the estimation of the total mass eruption rate, the fraction of airborne 'fine ash', the spatial distribution of ash, and the particle size distribution of the ash at the source. The uncertainty arises because the physical processes near the source are either poorly understood or require more complex and computationally demanding modelling [3] than what is available in the short time frames required for operational volcanic ash forecasting. Temporal variations of the source term, which are mainly driven by internal volcanological processes, are another source of uncertainty. Apart from these source-term uncertainties, all forecasts of ash are plagued by uncertainties in the ash transport and removal mechanisms within dispersion models. These include errors in the NWP fields [4–6], and parameterization of various physical processes such as sedimentation, deposition, and aggregation.

A commonly used approach to estimating mass eruption rates is based on the work of Mastin et al. [7]. In that study, data collected from nearly 30 past eruptions were used to estimate a power-law relationship between mass eruption rate and eruption height. The mass eruption rates in the case studies were estimated using field studies of volcanic tephra deposits. The empirical mass eruption rate was demonstrated to follow an approximate fourth-power relationship with the eruption height, in close agreement with theoretical predictions based on plume-rise modelling [8–10]. Many VAACs use such empirical relationships to estimate mass eruption rates [11]. Despite the good agreement with theory, the errors associated with these estimates can be significant, typically a factor of four [7]. In addition, these relationships predict the total mass eruption rate but do not specify the amount of mass in the size ranges considered most relevant for aviation forecasting, typically less than about 100 μm in diameter. Hence, the estimated mass eruption rate must be multiplied by a parameter, commonly called the 'fine ash fraction', to specify this amount. Mastin et al. [7] proposed a classification scheme to assign the fine ash fraction to different volcanos based on field studies of these or similar volcanoes during previous eruptions, but this approach is crude at best and does not work very well in general. Many VAACs instead use a set value for the fine-ash fraction, typically 5% [2]. Apart from the difficulties outlined above, the Mastin et al. [7] approach does not predict the correct spatial distribution of mass. In the absence of this information, the commonly used approach is to assume a uniform distribution of mass in the vertical. In addition, the particle size distribution needs to be specified and this is usually based on post-event analyses of previous eruption case studies.

A different approach towards an improved source term and reliable quantitative forecasts is to make use of satellite data. Over the last 10 years, the widespread availability of better multispectral sensors on geostationary satellite platforms like Himawari-8 has spurred the development of sophisticated remote-sensing algorithms for volcanic ash [12–14]. These algorithms, in addition to being able to better detect volcanic ash, enable the retrieval of volcanic ash cloud properties such as cloud top height, mass load and particle size information. This quantitative ash cloud data, particularly the mass load, combined with various inverse modelling methods [15–23] enables the estimation of the source mass emission rate, including the appropriate vertical distribution of mass, that needs to be specified for dispersion models. In this approach, the source mass emission rate does not require multiplication by the fine ash fraction because the retrieved mass loads are predominantly associated with the fine ash component and not the whole spectrum of particle sizes. This becomes clear if we consider that the retrievals have a typical upper effective radius cut-off of about 15 μm [14], which implies that particles less than 50 μm in radius (or equivalently, 100 μm in diameter) contain about 90% of the total mass within a retrieved pixel at this upper retrieval limit, assuming a log-normal number distribution with a geometric standard deviation of 2.1; the mass proportion in this particle size range will be even higher when the effective radius is significantly less than the retrieval limit. It should be noted that this is the distal fine ash fraction. Even though there might be higher proportions of larger particles closer to the source, these will fall out quickly and are, therefore, less important for longer-range aviation forecasting. Therefore, in this paper, any reference to fine ash fraction as derived from satellite retrievals is to be understood in this sense. A drawback of this approach is that the remote-sensing algorithms are not always able to successfully detect and retrieve ash properties due to surrounding

meteorological (ice and water) clouds. This problem is particularly acute in the Darwin VAAC region due to the frequent presence of convective clouds.

In this paper, we present a new approach that addresses limitations in the empirical and satellite-based approaches. The satellite-based approach has the advantage of not requiring knowledge of the ash mass fraction, but it has the disadvantage of being dependent on atmospheric conditions (such as presence of meteorological cloud cover) for its usability in operational contexts. The empirical approach, in addition to other problems outlined above, requires specification of the poorly understood fine ash mass fraction, but it has the advantage of being usable regardless of the atmospheric conditions. The new approach uses satellite retrievals obtained from the NOAA VOLcanic Cloud Analysis Toolkit (VOLCAT) software, based on the algorithm of Pavolonis et al. [14], with inverse modelling methods based on multidimensional sampling of model source parameter space [3,6,24–26], to estimate the source term (total mass emission rate, spatial mass distribution, and particle size distribution) for 14 eruption case studies in the Darwin VAAC area. These source term quantities are expressed as relatively simple functions of a few parameters such as the fine ash fraction, umbrella cloud diameter, and mean particle radius. From these cases, we then estimate empirical relationships between the calculated source-term parameters and eruption height as in the approach of Mastin et al. [7]. In this way, we aim to develop a system that is usable irrespective of atmospheric conditions but is less dependent on ad hoc parameters. In addition, because several source parameters become functions of eruption height, in this formulation the dimensionality of the inverse modelling problem is greatly reduced, thereby making it better suited for operational requirements.

## 2. Methodology

### 2.1. Ash Observations

Automated detection and retrieval of volcanic ash are obtained using the VOLCAT remote-sensing software. VOLCAT provides both automated detection [27] of ash in satellite imagery as well as retrieval of ash properties such as mass load for the detected pixels [14]. As noted in Section 1, the algorithm performs retrieval for pixels with ash particle effective radii less than 15 μm. However, as discussed by Pavolonis et al. [14], this should not be taken to imply that larger particles do not contribute to the radiance measured by the satellite. The particle size distribution is assumed to be log-normal (with a geometric standard deviation of 2.1), and an ash cloud with an effective radius at the upper limit of 15 μm, for example, will have a significant portion (~60%) of its mass contributed by particles larger than 15 μm. Retrievals from 14 eruptive events (listed in Table 1), all within the tropics (13 in the Darwin VAAC region and one in the Wellington VAAC region), are used as the base data to enable the estimation of the relationships between various source parameters and eruption height. Part of this dataset is also used for forecast verification purposes. These events occurred between 2014 and 2019 and represent eruptions with top heights extending from the mid-troposphere to the lower stratosphere. Events from July 2015 onward are sampled using the JMA Himawari-8 satellite; prior to that date, the older, lower-resolution MTSAT-2 was used. Additional discussion of the case studies is provided in Section 3. VOLCAT does not have a 100% success rate for detection of ash for a variety of reasons, including substantial meteorological cloud cover [28]. However, we have specifically chosen case studies with successful detections of ash over most of the cloud within a time window of at least a few hours in this study.

**Table 1.** Case studies arranged by decreasing eruption height. Also shown are estimated eruption start times (in the format year/month/day/hour) and time windows (relative to the start time) used in multidimensional inverse modelling (see Section 4), low-dimensional data assimilation (see Section 5), and forecast verification (see Section 5). For Rinjani I and II, the eruption is continuous, so the start time is somewhat arbitrary, as explained in more detail in Section 3. The roman numbers indicate that different eruption phases of the same volcano and S. Api is an abbreviation for Sangeang Api.

| Case Study Index | Volcano | Approximate Start Time (Date/UTC) | Multidimensional Inverse Modelling Time Window (Hours Relative to Start Time) | Data AssimilationAnalysis Time Window (Hours Relative to Start Time) | Forecast Verification Time Window (Hours Relative to Start Time) |
|---|---|---|---|---|---|
| 1 | Kelut | 2014/02/13/1615 | 7.25–10.25 | 7.25–10.25 | 11.25–16.25 |
| 2 | S. Api I | 2014/05/30/0800 | 5.5 | 5.5 | 6.5–11.5 |
| 3 | Manam I | 2015/07/31/0130 | 0.5–3.0 | 0.5–1.0 | 1.33–3.33 |
| 4 | Tinakula | 2017/10/20/2330 | 0.5–2.5 | 0.5–2.0 | 2.33–4.17 |
| 5 | S. Api II | 2014/05/30/1700 | 1.5–2.5 | 1.5–2.5 | 3.5–8.5 |
| 6 | Manam II | 2018/12/08/0300 | 1.83–4.67 | 1.83–2.67 | 3.67–4.5 |
| 7 | Soputan I | 2016/01/04/2240 | 0.33–2.0 | 0.33–0.83 | 1.0–2.0 |
| 8 | Soputan II | 2016/01/05/0600 | 1.17 | 1.17 | 1.33–2.67 |
| 9 | S. Api III | 2014/05/30/2000 | 0.5–2.5 | 0.5–2.5 | 3.5–8.5 |
| 10 | Merapi | 2018/05/11/0030 | 1.0–4.0 | 1.0–3.0 | 3.5–5.5 |
| 11 | Rinjani III | 2016/08/01/0300 | 1.17–3.33 | 1.17–2.0 | 2.83–4.0 |
| 12 | Agung | 2019/05/24/1130 | 0.5–3.0 | 0.5–2.5 | 3.0–5.0 |
| 13 | Rinjani I | 2015/11/04/1200 | 18.5–23.5 | 18.5–21.5 | 22.5–27.5 |
| 14 | Rinjani II | 2015/11/05/0000 | 16.5–19.5 | 16.5–19.5 | 20.5–25.5 |

## 2.2. Dispersion Model

The hybrid single particle Lagrangian integrated trajectory (HYSPLIT) model [29,30] is used to simulate the dispersion of ash from the source. In this study, HYSPLIT was driven by an ensemble of 24 NWP fields from the ACCESS-GE global model [31], which has a timestep of 3 h and a resolution of about 60 km in the tropics. As well as dispersion by gridded wind fields and parameterised turbulence, the bureau's version of HYSPLIT has a dry deposition scheme which removes particles from the atmosphere by sedimentation, and eventual surface deposition, using the Ganser fall speed formulation [32] with a shape factor of 0.8 and a wet deposition scheme.

## 2.3. Source Term in Reference Runs

The source term is a cylinder of particles, $D$ in diameter, with its base at the volcano summit, $H_0$, and top (eruption height), $H$, initiated at time $t_0$ for a duration $\tau$. In the standard setting, which is used to provide reference data by which we evaluate model improvements in the rest of this paper, $D$ is set to 20 km, $H$ is determined mainly by VAAC reports of the eruption height; $t_0$ and $\tau$ are estimated by a combination of VAAC reports and manual inspection of satellite data. The mass emission rate of fine ash (in kg s$^{-1}$) is determined by the empirical Mastin relationship $\dot{M}_f = \epsilon \dot{M}$, where $\dot{M} = 140.8(H - H_0)^{4.15}$ (with $H$ and $H_0$ in km) and $\epsilon$ is the fine-ash fraction, which is specified to be 5% [2]. The distribution of mass within the eruption column is assumed to be uniform. The particle size distribution is based on the distribution obtained by Hobbs et al. (1991) [33] and comprises a continuous spectrum of particle radii, $r$, in the range $0 < r < 50$ µm, which defines the particle size range considered most relevant for aviation forecasting. The emission rate is assumed to be a step function in time; it is constant for a duration $\tau$ and zero thereafter. This simple source formulation is similar to what is currently used at the Darwin VAAC and in many other volcanic ash advisory centres.

## 2.4. Source Term in Experimental Runs

In the experimental runs, which are the focus of this paper, we introduce a crude representation of an umbrella cloud. The source is a still cylinder of particles; however, the diameter, $D$, of the cylinder represents the size of the umbrella cloud at the time that wind-driven dispersion becomes the dominant ash transport mechanism. The initialisation time is at $t_0 + \Delta t_0$ rather than the start of the eruption, $t_0$. The time lag $\Delta t_0$ is the time taken for the umbrella cloud to reach a diameter of $D$. Prior to the

initialisation time, it is assumed that the umbrella cloud grows axisymmetrically at a rate determined by $\Delta t_0$ and $D$. In general, the diameter of the ash column is less than the diameter of the umbrella cloud and should be initialised at $t_0$ rather than $t_0 + \Delta t_0$, but here we choose to ignore this distinction to minimise the number of free parameters and errors arising from initialising the dispersion model from a small area due to NWP field errors. The emission duration is $\tau$, relative to $t_0 + \Delta t_0$ rather than $t_0$ as in the reference runs.

The source mass emission rate of fine ash particles for a given height $h$ and particle radius $r$ can be written as:

$$\dot{m}(h, r) = \epsilon f(h) g(r) \dot{M},$$ (1)

where $f(h)$ in Equation (1) describes the vertical mass distribution:

$$f(h) = \frac{c + \exp\left[-\frac{(h - \mu_h)^2}{2\sigma_h^2}\right]}{\int_{H_0}^{H}\left\{c + \exp\left[-\frac{(h - \mu_h)^2}{2\sigma_h^2}\right]\right\}dh},$$ (2)

which comprises a uniform emission rate denoted by $c$, describing the eruption column, and a variable emission rate describing the umbrella cloud, which is chosen to be Gaussian with centre $\mu_h$ and standard deviation $\sigma_h$. In Equation (2), we define the width of the umbrella cloud $\delta = 4.29193\,\sigma_h$, which corresponds to full width at one-tenth maximum of the Gaussian distribution. For convenience, we also define the ratio of the mass emitted from the umbrella cloud to the mass emitted from the eruption column as $\beta = \sqrt{2\pi}\sigma_h / c(H - H_0)$ and deal with $\beta$ rather than $c$ hereon. This expression can be derived by integrating in the vertical to obtain the total mass emission rate for both the umbrella cloud and eruption column and dividing the two expressions, assuming that $\sigma_h << H - \mu_h$ and $\sigma_h << \mu_h - H_0$. This interpretation of $\beta$ strictly only applies to the larger eruptions considered in this study. For smaller eruptions, which do not satisfy this condition, the mass ratio is in general very small as most of the mass is contained within the eruption column, and $\beta$ will be an overestimate of the mass ratio.

In addition to improving the umbrella cloud representation of the source, in the experimental runs the particle size number distribution is assumed to be described by a log-normal distribution with a variable mean geometric radius, $\mu_r$. The particle size mass distribution is given by:

$$g(r) = \frac{r^2 \exp\left[-\frac{(\ln r - \ln \mu_r)^2}{2(\ln \sigma_r)^2}\right]}{\int_0^R\left\{r^2 \exp\left[-\frac{(\ln r - \ln \mu_r)^2}{2(\ln \sigma_r)^2}\right]\right\}dr},$$ (3)

where in Equation (3) $\sigma_r$ is the geometric standard deviation, which is fixed to a value of 1.5, and R = 50 μm is the maximum particle radius retained in the distribution [3]. The total mass eruption rate, $\dot{M}$, in Equation (1) is given by the Mastin relationship as in the reference runs.

In summary, the parameters required to construct the improved source term are $H$, $\Delta t_0$, $D$, $\tau$, $\mu_h$, $\delta$, $\beta$, $\mu_r$ and $\epsilon$. In the following section we describe how this can be done by inverse modelling.

2.4.1. Inverse Modelling by Multidimensional Optimisation

The above source parameters can be estimated for given observations of ash by using the inverse modelling approach of Zidikheri et al. [3,6,24–26]. The method is based on sampling the uncertainty bound of each parameter to form a large parameter grid (in hyperspace) given an initial estimate of the parameter bounds. Each point in this parameter grid (except for $\epsilon$, as further discussed below) represents a possible configuration of the dispersion model source term, and each of these possible source configurations is used to run the dispersion model forward in time. The resulting model output is compared against observations within a specified time window (determined by the availability of observations) by using the pattern correlation metric [24]. We refer to these simulations as trial simulations in this paper. The inverse modelling algorithm then selects the best-performing trial

simulations to form the forecast ensemble. The optimal pattern correlation cut-off, above which a given trial simulation is included in the forecast ensemble, is determined by choosing a range of different cut-offs and selecting the cut-off that maximises the Brier skill score ([6] and references therein) of the forecast ensemble within the analysis time window.

Because the inverse modelling method relies on a grid search algorithm, the number of forward trial runs grows exponentially with the number of free parameters, and this limits its ability to handle the number of parameters required to completely solve the inverse problem posed above. This is exacerbated by using an NWP ensemble—such as ACCESS-GE—to drive the dispersion model because each source parameter configuration needs to be run with all the meteorological ensemble members. In this study, we ameliorate these problems by firstly limiting the number of grid points per parameter and secondly by dividing the problem into two parts, namely, the detection problem and the retrieval problem.

The detection problem involves optimising the parameters relative to the ash detections obtained from VOLCAT. In practice, this simply means transforming the VOLCAT mass load retrievals into a new binary-valued field, which has values of 1 where the mass load is greater than zero (and 0 elsewhere). The mass loads from the trial simulations are similarly given a value of 1 where there is any ash and 0 elsewhere to create a new binary-valued simulated detection field. The source parameters are then optimised with respect to these binary-valued fields. This approach has been used successfully in the past to estimate parameters such as eruption height [24–26]. Using this approach, we can initially disregard the source parameters that describe the total mass eruption rate and its vertical distribution, namely $\epsilon$, $\mu_h$, $\mu_r$, $\delta$, and $\beta$, because they do not affect the simulated ash detection field. These parameters only change the amount of ash at a given location. The only parameters that affect the location of simulated ash are $H$, $\Delta t_o$, $D$, and $\tau$ (as well as the meteorological ensemble members). Having used the ash detections to reduce the number of possible grid points for the inverse problem, the remaining grid points (optimal with respect to detections) are then combined with the grid points formed by sampling the parameters $\mu_h$, $\mu_r$, $\delta$, and $\beta$ (the parameter $\epsilon$ is computed as described further below). This new parameter grid is then optimised relative to ash mass load retrievals; this forms the so-called retrieval problem. Splitting the problem this way is computationally more efficient because we do not have to sample the whole parameter space spanned by the unknown parameters.

The ash mass that is retrieved from satellite data is only a fraction of the total mass emitted from the volcano because it is only the fine ash that remains in the atmosphere at the time scales of hours or longer and can be detected by remote-sensing algorithms. This detectable component is given by $\dot{M}_f = \epsilon \dot{M}$, where $\dot{M}$ is the total mass eruption rate as estimated from the Mastin et al. [7] empirical relationship, and $\epsilon$ is the fine ash fraction, which is determined by the inverse modelling procedure. Note, however, that $\epsilon$ is just a scaling factor and has no effect on the pattern correlation. It is computed by scaling each of the trial simulations so that they have on average the right magnitudes relative to the ash mass load retrievals; that is:

$$\epsilon = \frac{1}{N_t} \sum_{i=1}^{N_t} \frac{\overline{\Lambda_o}(t_i)}{\overline{\Lambda_s}(t_i)}, \tag{4}$$

where $\overline{\Lambda_o}(t_i)$ is the spatial mean of the observed (retrieved) mass loads at time $t_i$, $\overline{\Lambda_s}(t_i)$ is the spatial mean of the trial simulation mass loads, and $N_t$ is the number of temporal observations within the analysis time window. It should be noted that defining the fine ash fraction in this way in Equation (4) implicitly assumes that the Mastin relationship is accurate. If the Mastin relationship is in error, then $\epsilon$ accounts for both the fine ash fraction and the error in the Mastin relationship.

2.4.2. Deriving Empirical Scaling Relationships between Source Parameters and Eruption Height

The inverse modelling algorithm as described above requires the sampling of a nine-dimensional parameter grid (eight source parameters plus the space formed by the meteorological ensemble members). This generally requires several thousands of trial simulations and is, therefore, not practical

for operational volcanic ash forecasting, where timely forecasts are essential. In addition, the availability of satellite retrievals is strongly dependent on atmospheric conditions such as the presence of (water and ice) clouds and so, on the face of it, the method is not very reliable for operational use, especially in the tropical region covered by the Darwin VAAC. Therefore, we firstly seek to find empirical relationships between various parameters and eruption height using past eruption case studies. Having established the approximate dependence of the source parameters to eruption height, the optimisation problem is reduced to finding optimal eruption heights and NWP ensemble members only, which is much less computationally expensive, and therefore suitable for operational requirements. In addition, because eruption height can be estimated even without satellite retrievals [25,26], the method could still be used even when no real-time satellite retrievals are available.

Motivated by the known approximate fourth-power relationship between mass eruption rate and height (relative to the summit), we seek similar power-law relationships for the other source term parameters (except $\tau$, for reasons explained below) as follows:

$$y_i^{jk} = a_i \varepsilon_i^{jk} \left( H^{jk} - H_0^j \right)^{b_i}, \tag{5}$$

where $y_i^{jk}$ in Equation (5) is the $i$-th element of the vector $\boldsymbol{y} = \left( \Delta t_0^{jk}, \ D^{jk}, \ \mu_h^{jk}, \ \delta^{jk}, \ \beta^{jk}, \ \epsilon^{jk}, \mu_r^{jk} \right)$, comprised of the different source parameters; $\varepsilon_i^{jk}$ is a multiplicative error factor, which arises both due to sampling error and the fact that a simple power-law relationship between model parameters and eruption height is only a crude approximation at best (the error is considered as multiplicative, rather than additive, because some parameters, such as mass eruption rate, span several orders of magnitude and, therefore, the errors should scale accordingly); the superscript $j$ represents different case studies, $N_S$ in number ($j = 1, 2, 3 \ldots N_S$), obtained by considering a range of volcanic eruptions chosen to span a wide range of maximum eruption heights; the superscript $k$ represents different forecast ensemble members ($k = 1, 2, 3 \ldots N_M^j$, where $N_M^j$ is the total number of forecast ensemble members yielded by the inverse model, and is different for each case study, $j$); $a_i$ and $b_i$ are unknown coefficients. We have not included the eruption duration, $\tau$, in the vector of source parameters above and instead assume it is an independent variable whose value must be estimated by inspection of satellite imagery, or other means, at the time of a volcanic eruption. The coefficients $a_i$ and $b_i$ are calculated by first converting (5) into logarithmic form,

$$Y_i^{jk} = A_i + B_i X^{jk} + E_i^{jk}, \tag{6}$$

where $X^{jk} = \log_{10} \left( H^{jk} - H_0^j \right)$, $Y_i^{jk} = \log_{10} y_i^{jk}$, $A_i = \log_{10} a_i$, $B_i = b_i$ and $E_i^{jk} = \log_{10} \varepsilon_i^{jk}$. The coefficients $A_i$ and $B_i$ may be evaluated by linear regression (that is, minimisation of $\sum_{j=1}^{N_s} \sum_{k=1}^{N_m^j} E_i^{jk} E_i^{jk}$). However, to avoid the problem of dealing with data points with errors that might not be completely uncorrelated (due to the presence of the clusters of data points from the same case study), we first calculate the ensemble mean of each term in (6):

$$\langle Y_i^j \rangle = A_i + B_i X^j + E_i^j, \tag{7}$$

where $\langle Y_i^j \rangle = \frac{1}{N_M^j} \sum_{k=1}^{N_M^j} Y_i^{jk}$, $\langle X^j \rangle = \frac{1}{N_M^j} \sum_{k=1}^{N_M^j} X^{jk}$, and $\langle E_i^j \rangle = \frac{1}{N_M^j} \sum_{k=1}^{N_M^j} E_i^{jk}$. The error, $\langle E_i^j \rangle$, in Equation (7) is then less likely to be significantly correlated for different values of $j$ (for a given parameter, $i$), as each case study ($j$) utilises different observations of mass load. Prior to regressing the data points ($\langle X_i^j \rangle$, $\langle Y_i^j \rangle$), we also need to consider that the data points are associated with different degrees of uncertainty for the different case studies because each case study is associated with different numbers of model ensemble members, degree of agreement with observations, and observation times relative to the initialisation time. Therefore, it is appropriate to first weight the data points so that:

$$\widetilde{Y}_i^j = A_i w_i^j + B_i \widetilde{X}^j + \widetilde{E}_i^j, \tag{8}$$

where in Equation (8) $\widetilde{X}^j = w_i^j \langle X^j \rangle$, $\widetilde{Y}_i^j = w_i^j \langle Y_i^j \rangle$, and $E_i^j = w_i^j \langle E_i^j \rangle$. The weights are prescribed as:

$$w_i^j = \frac{\langle r^j \rangle (\overline{t_a^j} - t_0)}{\Omega_i^j} \eta_i^j.$$

(9)

Each weight, as indicated in Equation (9), is directly proportional to the product of the ensemble mean pattern correlation value, $\langle r^j \rangle$, and the mean analysis time, $\overline{t_a^j}$, relative to the eruption start time $t_0$. This ensures that parameter values associated with high pattern correlation values receive greater weight in the regression. Furthermore, it ensures that case studies with high pattern correlations obtained from using observations at later times, relative to the start time, receive greater weight than case studies utilising only observations soon after the eruption start time. The reasoning behind this is that pattern correlation values typically fall with increasing time as the dispersion pattern usually becomes more complex with increasing time; this must be accounted for when regressing data points based on observations at different times relative to the start time. Each weight is also inversely proportional to the ensemble standard deviation $\Omega_i^j$ from the mean parameter value $\langle Y_i^j \rangle$, which is another measure of the degree of uncertainty of the inversion. This ensures that mean parameter values associated with small standard deviations receive greater weights than parameter values with large standard deviations. The coefficient $\eta_i^j \in \{0, 1\}$ is a binary variable. It enables some case studies ($j$) to be removed entirely from the regression for certain parameters ($i$) (by setting $\eta_i^j = 0$). This point will be discussed in more detail in Section 4.

The parameters $A_i$ and $B_i$ are estimated by linear regression from Equation (8), as shown in Appendix A. Although we estimate the parameters by linear regression of the weighted ensemble means, it is also useful to know the statistics of the errors in the ensemble members due to linear regression of the ensemble means (the statistics of the parameter $\varepsilon_i^{jk}$ in Equation (5)). These are described by the error covariance matrix, $C$, whose elements are given by:

$$C_{ii'} = \sum_{j=1}^{N_S} \sum_{k=1}^{N_M^j} W_{ii'}^{jk} E_i^{jk} E_{i'}^{jk},$$

(10)

where in Equation (10)

$$W_{ii'}^{jk} = \frac{w_i^j w_{i'}^j}{\sum_{j=1}^{N_S} \sum_{k=1}^{N_M^j} w_i^j w_{i'}^j}.$$

(11)

The diagonal elements of $C$ contain the variances of the parameter errors, so the standard multiplicative errors (in normal space, not logarithmic) for each parameter are given by:

$$\sigma_i = 10^{\sqrt{C_{ii}}}.$$

(12)

## 3. Application of Multidimensional Inverse Modelling to Selected Case Studies

The eruption case studies that are used in this paper for the purpose of estimating relationships between various source parameters and eruption height, as outlined in Section 2, are listed in Table 1. These case studies are discussed more comprehensively in the Supplementary Materials. Table 1 also lists the start times and observational time windows used in the multidimensional inverse modelling procedure for estimating parameter values. In this section, we shall also compare the performance of the reference runs, which employ a simple source formulation without source optimisation, with experimental runs. The experimental runs employ a more complicated source (with a crude representation of umbrella cloud effects) that is optimised with respect to VOLCAT satellite retrievals; these results are summarised in Table 2.

**Table 2.** Case studies arranged by decreasing eruption height. The a priori height estimates are mostly based on Volcanic Ash Advisory Centre (VAAC) reports. For the cases where the VAAC estimates are significantly different to the values inferred by inverse modelling, we have indicated an alternate value in parenthesis. The alternate value is based on CALIPSO imagery for Kelut (Case Study 1) and pilot reports for S. Api I and Rinjani III; the rest are based on brightness temperature analyses. All heights are relative to mean sea level. The mass loads shown are approximate peak values that we have extracted from the plots of retrieved and simulated values. In the latter, we have removed any grid points with particle effective radii greater than 15 μm, the upper limit of the retrievals, in order to facilitate a better comparison between the two.

| Case Study Index | Volcano | A Priori Height Estimate (km) | Inverse Model Height Estimate (km) | VOLCAT Mass Load (g m$^{-2}$) | Reference Run Mass Load (g m$^{-2}$) | Optimised Mass Load (g m$^{-2}$) |
|---|---|---|---|---|---|---|
| 1 | Kelut | 16.7 (26) | 25.3 | 40–120 | 800–1200 | 60–70 |
| 2 | S. Api I | 15.2(~20) | 20.8 | 15 | 45 | 20 |
| 3 | Manam I | 20 | 20.1 | 60–200 | 3000–7000 | 200–300 |
| 4 | Tinakula | 10.7(~17) | 18.6 | 150 | 900–3000 | 80–160 |
| 5 | S. Api II | 15.2 | 15.9 | 14–18 | 250–500 | 15–23 |
| 6 | Manam II | 13.7 | 14.3 | 23–47 | 700–1500 | 50–80 |
| 7 | Soputan I | 12.8 | 13.9 | 7–11 | 900 | 10–30 |
| 8 | Soputan II | 12.5 | 14.0 | 10 | 1100 | 15 |
| 9 | S. Api III | 15.2 | 13.5 | 12–15 | 300–900 | 11–17 |
| 10 | Merapi | 15.2 (~8) | 7.8 | 4–9 | 70–90 | 4–6 |
| 11 | Rinjani III | 6.1(9.8) | 8.5 | 4–7 | 1–3 | 4–6 |
| 12 | Agung | 4.6 (6–7) | 7.0 | 3–6 | 6–15 | 3–5 |
| 13 | Rinjani I | 6.1 | 6.2 | 3–5 | 3 | 3 |
| 14 | Rinjani II | 6.1 | 6.0 | 2–3 | 3 | 3–4 |

The results in Table 2 show that the eruption height estimates based on inverse modelling agree quite well with Darwin VAAC estimates in most cases, albeit with a slight tendency towards higher values. In cases where there are significant discrepancies, further analysis using Cloud-Aerosol Lidar and Infrared Pathfinder Satellite Observation (CALIPSO) overpasses (Kelut), pilot reports (Sangeang Api I, Rinjani III), and satellite brightness temperature imagery (Tinakula, Merapi, and Agung) reveals that the correct eruption height is likely to be closer to the inverse modelling estimates. Table 2 also reveals that the use of inverse modelling improves the simulated mass load estimates. In the reference runs, the mass loads are significantly higher than those in the retrievals, in some cases by more than an order of magnitude. This is especially true for the stronger (higher level) eruptions. This phenomenon was also noted by Zidikheri et al. (2017) [3].

The application of multidimensional inverse modelling to all the 14 eruption case studies reveals significant improvement to forecast performance relative to reference runs without optimised source parameters using metrics such as mass load values, pattern correlations, and Brier skill scores. However, as noted in Section 2, because the number of trial simulations grows exponentially with the number of source parameters, it is very computationally intensive and therefore not suitable for implementation in operational contexts. We instead use the inferred parameter values to estimate relationships between various parameters and eruption height, which is done in Section 4, with the aim of constructing more computationally efficient data assimilation schemes, as demonstrated in Section 5.

## 4. Estimating Relationships between Source Parameters and Eruption Heights

Here we focus on representing the source term parameters obtained in the inverse modelling procedure, described in Sections 2 and 3, as power-law relationships with respect to eruption height above summit, $H - H_0$. The coefficients $a$ and $b$ defining the power-law $a(H - H_0)^b$ for each parameter are obtained by linear regression of the (log-transformed) data points as described in Section 2, and are shown in Table 3. The variation of the parameter values with eruption height are shown in Figures 1 and 2. The empirically derived curves (solid blue) and the error margins (dashed and dotted blue) are shown in both figures. It should be noted that the error margins are based on Equation (12). They describe the average deviations of the parameter ensemble members relative to the optimal

fits, and not just the deviations of the ensemble means from the optimal fits, which are smaller in magnitude. The weights employed for each data point (Equation (9)) are shown in red. As alluded to in Section 2, some data points have been assigned a weight of 0 for some parameters (by setting $\eta_i^j = 0$ in Equation (9)). These include the parameters $\Delta t_0$ and $D$ in Rinjani I and II because in those case studies the simulations were initialised within a time interval of 24 h prior to the assimilation time window indicated in Table 1 rather than the start of the eruption (as further discussed in the Supplementary Section) and, therefore, $\Delta t_0$ and $D$ in the sense of Section 2 could not be determined. In addition, the data points for the parameters $\mu_h$, $\delta$, and $\beta$, which are related to the vertical mass distribution, have also been assigned weights of 0 for small eruptions with top heights less than about 5 km above the summit because this height range was deemed too small to accurately determine the correct vertical distribution of mass.

**Table 3.** Source parameters determined by empirical power-law relationships in this study. The top four data rows show the power law coefficients *a* and *b* obtained for different source parameters and their respective standard regression errors. The bottom seven data rows show the coefficients of the covariance matrix *C* as calculated in Equation (10) with the diagonal elements shown in bold. The units of the source parameters are also indicated ($\epsilon$ and $\beta$ are dimensionless ratios and $\mu_h$ is in km above the summit). The independent variable $H - H_0$ in the power-law relationships is measured in km.

| Parameter Name | Start Time Lag | Diameter | Fine Ash Fraction | Umbrella Elevation | Umbrella Depth | Mass Ratio | Mean Particle Radius |
|---|---|---|---|---|---|---|---|
| Symbol (unit) | $\Delta t_0$ (min) | $D$ (km) | $\epsilon$ | $\mu_h$ (km) | $\delta$ (km) | $\beta$ | $\mu_r$ (µm) |
| *a* | 3.8213 | 4.7635 | 0.0747 | 2.2227 | 94.1063 | 0.6866 | 0.3038 |
| *b* | 0.6989 | 0.9520 | −1.0245 | 0.5905 | −1.5062 | 0.9391 | 0.5904 |
| $\delta a$ | 2.8273 | 1.4567 | 1.7920 | 1.5340 | 8.9545 | 2.6550 | 2.2683 |
| $\delta b$ | 0.3648 | 0.1440 | 0.2321 | 0.1365 | 0.7249 | 0.3233 | 0.2885 |
| $\Delta t_0$ | 0.0252 | 0.0048 | 0.0200 | 0.0024 | −0.0195 | 0.0104 | −0.0086 |
| $D$ | 0.0048 | 0.0224 | 0.0326 | 0.0008 | 0.0036 | 0.0020 | 0.0044 |
| $\epsilon$ | 0.0200 | 0.0326 | 0.1459 | 0.0051 | 0.0217 | −0.0014 | −0.0045 |
| $\mu_h$ | 0.0024 | 0.0008 | 0.0051 | 0.0012 | −0.0058 | −0.0002 | 0.0056 |
| $\delta$ | −0.0195 | 0.0036 | 0.0217 | −0.0058 | 0.6889 | 0.0512 | 0.1015 |
| $\beta$ | 0.0104 | 0.0020 | −0.0140 | −0.0002 | 0.0512 | 0.6479 | 0.0338 |
| $\mu_r$ | −0.0086 | 0.0044 | −0.0045 | 0.0056 | 0.1015 | 0.0338 | 0.6720 |

(a)                                                    (b)

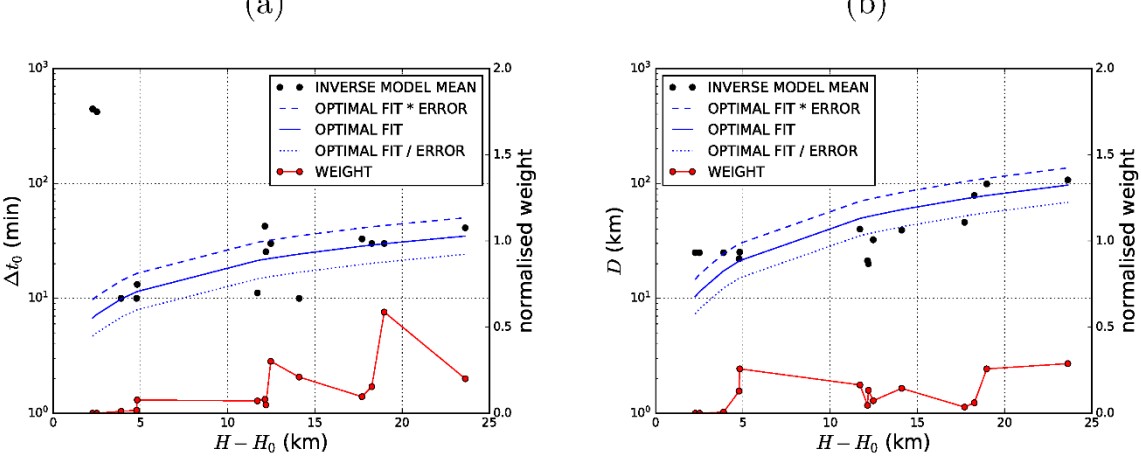

**Figure 1.** *Cont.*

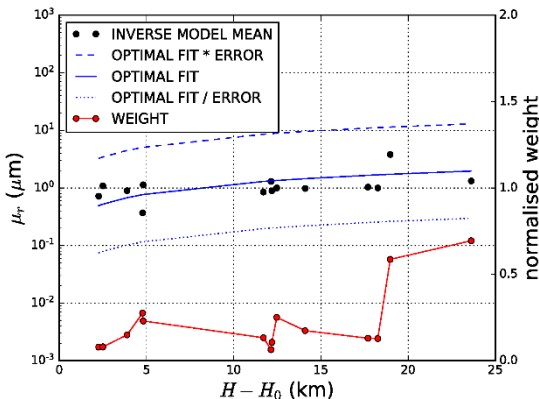

**Figure 1.** Empirically derived source parameters: (**a**) start-time time lag, $\Delta t_0$; (**b**) diameter, $D$; and (**c**) mean particle radius, $\mu_r$. Also shown are model-derived ensemble mean values (dots), optimal fit (solid blue), optimal fit multiplied by standard error factor (dashed blue), optimal fit divided by standard error factor (dotted blue) and prescribed normalised weights (solid red).

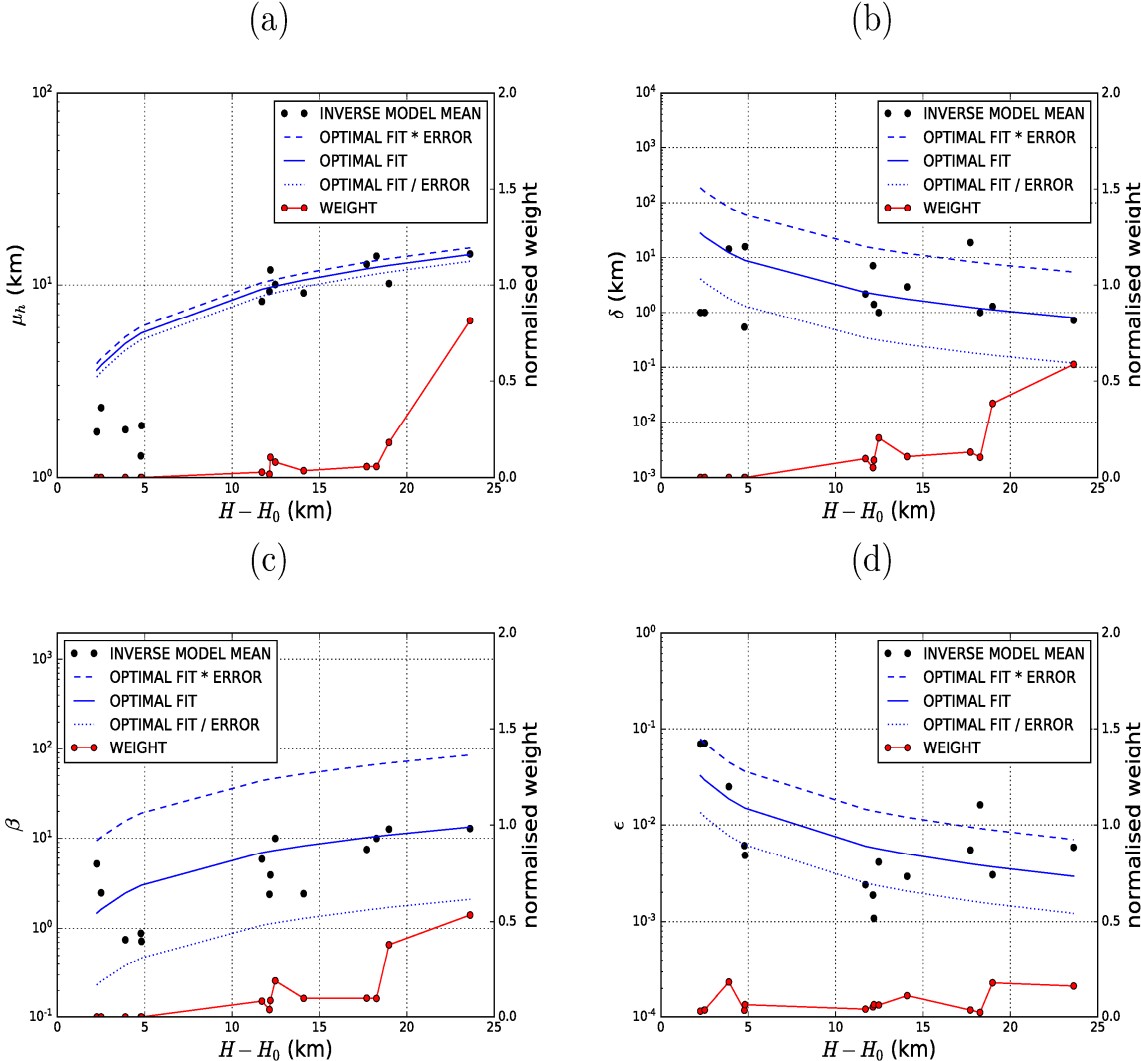

**Figure 2.** Same as in Figure 1 but for (**a**) umbrella centre, $\mu_h$; (**b**) umbrella depth, $\delta$; (**c**) umbrella to column mass ratio, $\beta$; and (**d**) fine-ash fraction, $\epsilon$.

The results in Table 3, Figure 1a, and Figure 1b show that the start-time time lag, $\Delta t_0$, grows slowly (power of ~0.7) with eruption height while the source diameter, $D$, grows approximately linearly (power of ~0.95) with eruption height. In a qualitative sense at least, both results appear to be consistent with gravity-current driven flows during the early stages of large eruptions. In the limit of small eruptions ($H \rightarrow H_0$), the parameterised $\Delta t_0$ and $D$ approach small values (3.8 min and 4.8 km, respectively), which are not significantly different from the settings used in many volcanic ash models (i.e., line source initialised at the eruption start time). The effects of the empirical power-law relationships for these two variables are, therefore, only significant for large eruptions in which both $\Delta t_0$ and $D$ are substantially larger than used in the default settings. For the largest eruption in this study (Kelut, 13 February 2014) the predicted start-time lag was over 30 min and the predicted umbrella cloud diameter nearly 100 km. The error margins are factors of 1.44 and 1.41 for $\Delta t_0$ and $D$, respectively, which indicates that the empirical relationship predicts the correct parameter values with reasonable skill.

The coefficients associated with the parameters describing the vertical distribution of mass at the source also appear to be consistent with the physical picture of umbrella clouds. The centre of mass of the umbrella cloud, $\mu_h$, grows with eruption height (albeit quite slowly, with a power of only ~0.59) while the depth of the umbrella cloud, $\delta$, falls rapidly (power of ~−1.5) with increasing eruption height. In addition, the ratio of mass in the umbrella cloud to the mass in the eruption column, $\beta$, grows approximately linearly with increasing eruption height. Therefore, the cumulative effect of these relationships is that for large eruptions, the vertical mass distribution is tightly confined within the altitudes spanned by the umbrella cloud while for small eruptions the distribution is approximately uniform (in the limit $H \rightarrow H_0$, $\delta \sim 94$ km, which is in effect a uniform distribution). As for $\Delta t_0$ and $D$, the effect of these new parameterisations, therefore, would only be significant for large eruptions. The error for $\mu_h$ is a factor of 1.08, which is smaller than for the other parameters, indicating relatively high predictive skill for correct parameter value (see Figure 2a). The errors for $\delta$ and $\beta$ are however large, being factors of 6.76 and 6.38, respectively, and therefore we expect limited predictive skill for these parameters using the empirical relationships (see Figure 2b,c). The large errors probably reflect the fact that the vertical mass distribution is much more complex than the simple Gaussian function representation that we have employed. They could also be related to the fact that, for the smaller eruptions, both large values of $\delta$ and small values of $\beta$ result in a nearly uniform distribution, which leads to considerable scatter in the values obtained by inverse modelling.

The results in Table 3 also indicate that the fine ash fraction, $\epsilon$, falls approximately linearly with eruption height. Unlike the five parameters described above, whose characteristics can at least qualitatively be linked to the growth of a gravity-wave driven umbrella cloud, it is not completely clear why this is the case. There are other studies that have hinted that there is indeed an inverse relationship between eruption height and fine ash fraction. For example, Tupper et al. [34] found by running a high-resolution three-dimensional plume model that tropical volcanic eruptions result in the ash plume rising higher than suggested by the mass eruption rate, but with lower fine ash content. Gouhier et al. [35] also found that very strong eruptions have relatively small fine ash content by a using satellite-based methodology for estimating the fine ash fraction. The derived empirical relationship indicates that for small eruptions the fine ash fraction approaches 7% while for large eruptions, e.g., reaching tropopause height or 16 km in altitude, it is less than 1%. These results are consistent with previous studies [3,6], which found that for large eruptions (such as 13 February 2014 Kelut), the mass loads were vastly overestimated upon employing the default value of 5% for the fine ash fraction. The error margin for $\epsilon$ (and therefore in the fine ash emission rate) is a factor of 2.41, which is quite large (Figure 2d). However, we should put this uncertainty in the context of the commonly used Mastin scheme [7]. The error in the Mastin relationship is at least a factor of 4 [7]. In addition, in the Mastin approach, one must also specify the fine ash fraction, which is associated with significant uncertainty [2]. The estimate of the fine ash emission rate in this approach is, therefore, a significant improvement over the Mastin scheme in this regard.

The physics behind the parameterisation of the mean particle radius, $\mu_r$, is also not clear. It appears to grow slowly (power of ~0.59) with eruption height. Given that most of the VOLCAT mass load retrievals continue for at most a few hours past the commencement of an eruption in most case studies, it is very difficult to extract a reliable signal related to particle size distribution from the mass load retrievals because it typically takes several hours for the smaller particles to fall significant distances. However, the 13 February 2014 Kelut eruption (the data point with the highest $H - H_0$), seems to be an exception, with a clear signature of particles falling to lower levels [3] and that together with the relatively large computed weight may explain the slight bias towards large particle sizes in the results shown in Table 3 (implied by the power of 0.59 variation with eruption height). The error is a factor of 6.60, which indicates that the predictive skill to be gained from this empirical relationship is limited, however (Figure 1c).

## 5. A Computationally Efficient Data Assimilation Scheme Based on Empirical Source Relationships

In the context of this study, data assimilation refers to the use of satellite data to optimise volcanic ash forecasts. It comprises an analysis phase in which the inverse model is used to find optimal model parameters within a relatively short time window in which observations are available (usually from the start of the eruption up to the time at which the forecast is issued), and a forecast phase in which the optimal parameters are used to propagate the models further forward in time. In both phases, an ensemble approach is pursued, with uncertainties in the source parameters and different ensemble NWP realisations being used to generate the ensemble members. A principal aim of this study is to investigate the viability of a more computationally tractable data assimilation scheme, which does not require high-dimensional optimisation (that is, many free parameters). Another principal aim is to investigate the viability of providing quantitative forecasts of ash mass load even when satellite retrievals are not available in real time. These aims seek to address two principal constraints in an operational environment, namely the requirement for timely forecasts to provide end users with sufficient time to act on available information, and the challenging tropical environment of the Darwin VAAC in which satellite retrievals are frequently not available due to the frequent presence of ice and water clouds.

The development of empirical relationships between various source parameters and eruption height in Section 4 is crucial because it enables the search space for model parameters to be greatly reduced when satellite retrievals are available in real time. Instead of independently searching for optimal parameters in nine-dimensional space, requiring thousands of trial simulations, the problem is reduced to finding the optimal eruption height, which requires only a few hundred trial simulations (in the order of 10 simulations for each of the 20–40 meteorological ensemble members available). In the case when satellite retrievals are not available, but ash can still be delineated from surrounding clouds, the eruption height may be estimated using inverse modelling as shown in other studies [24–26] and the empirical source term relationships used to provide quantitative forecasts of mass load (it should be noted that in contrast to Section 2, in which we treat any simulated ash as part of the detection field, here we impose a lower mass load limit to detections, which we take to be 0.1 g m$^{-2}$ in order to better account for the variety of situations in practice such as long-lived eruptions in which some of the ash might have dissipated to quite low levels and, therefore, not be visible in satellite imagery or be detectable by automated algorithms such as VOLCAT). Finally, even when ash cannot be delineated from surrounding clouds, provided that an estimate of eruption height is available by other means, such as ground observer or pilot reports, the empirical relationships may still be used to provide quantitative forecasts of ash mass load. We show in this section how these various estimates of mass load in different operational contexts compare with estimates of mass load in reference runs, which utilise default settings for the source term as currently used operationally.

Ideally, we should use new case studies for this verification exercise. Unfortunately, identifying case studies with good retrievals, particularly in the tropical atmosphere, is a manual, time-consuming,

process and it was not possible to pursue this approach in this study. Instead, we use the same case studies used in the training phase for forecast verification. However, here we divide the satellite retrieval data into two parts, namely analysis data and forecast verification data as shown in Table 1. The analysis data are for the most part the same data that were used in the training phase with multidimensional inverse modelling (although there are some differences in case studies with short time windows in which retrievals were available, as shown in Table 1). In this case, the analysis data are used to perform the two-dimensional optimisation of height and meteorological ensemble members instead of the computationally intensive nine-dimensional optimisation, with empirical source relationships being used to determine the other source parameters as functions of eruption height. On the other hand, the forecast verification data, for the most part, comprise retrievals that were deemed incomplete in some sense because they only covered a fraction of the ash cloud (the rest of the ash cloud did not yield retrievals either because the ash was too optically thin, or too optically thick, or was obscured by cloud, or some other reason) and, therefore, not suitable for the training phase. These data, which in all cases comprise retrievals at times greater than the times used for the analysis time window, are used for verification purposes instead. Hence, even though the same case studies are used here as in the training phase, for the most part the forecast verification segment of the data were not used in the training phase. It is quite likely, therefore, that the forecast performance here will generally be indicative of the performance of the (low-dimensional) system in general because of this division of the dataset.

In the analysis phase, we envisage three operational scenarios as far as observations are concerned, namely that full retrievals are available, only detections are available, and no retrievals or detections are available. In the first two scenarios, the eruption height and meteorological ensemble members are optimised with respect to available observations (retrievals and detections, respectively) while in the last scenario there is no optimisation and the a priori eruption heights in Table 2 (in brackets) are used instead to determine the source parameter values from the derived empirical relationships. We calculate Brier skill scores for each scenario as shown in Figure 3. The skill scores for the analysis phase are shown in Figure 3a while the forecast phase skill scores are shown in Figure 3b. In the forecast phase we use available retrievals to calculate the Brier skill scores in all three scenarios. Brier skill scores greater than zero imply improvement relative to reference runs (see Section 2). In a broad sense, we can see that there is overall improvement irrespective of which observation type is used. However, there are some differences between the runs, which are also evident. In the analysis phase, optimising with respect to the retrieved mass loads leads to superior results, which is not a surprising result since that is what the algorithm is designed to do. What is encouraging, however, is the extent to which optimising with respect to ash detections only leads to better agreement between simulated mass loads and retrievals; in fact, in a few cases, the detection-optimised results are better than the results with retrieval optimisation. Employing the empirical source parameter relationships without optimising the eruption height also leads to improvement in all but three of the 14 case studies. These improvements are more modest, however, even in the 11 cases where the Brier skill score is positive. In the forecast phase (in which the retrievals are used only for verification and not to optimise the simulations in any way), the retrieval-optimised runs are still overall superior in the majority of cases, but the detection-optimised runs are actually significantly better in three cases and not much worse in about four or five cases.

(a) (b)

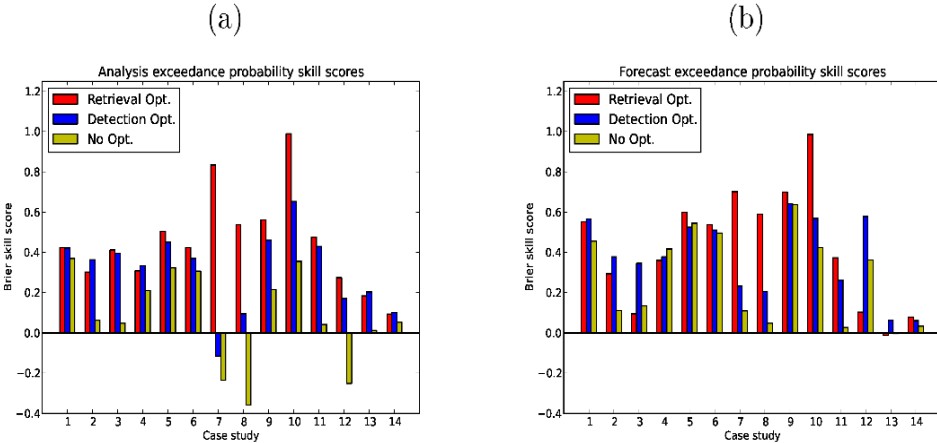

**Figure 3.** Brier skill scores for different case studies numbered from 1 to 14 (as listed in Table 1) for (**a**) analysis and (**b**) forecast phases in the low-dimensional optimisation runs. The results for the runs utilising retrievals are shown in red, the runs utilising detections only are shown in blue, and the runs without optimisation are shown in yellow (for which, in practice, there are no distinct analysis and forecast phases). Brier skill scores greater than zero imply improvement relative to reference simulations, which are chosen to be hybrid single particle Lagrangian integrated trajectory (HYSPLIT) runs with default source settings.

The runs not employing optimisation in the analysis phase (but employing empirical source relationships) are generally slightly poorer compared to the optimised runs in both phases. However, they perform better in the forecast phase than in the analysis phase. This is evident if we consider that in the analysis phase the non-optimised runs are inferior to the optimised runs in all cases while in the forecast case there are at least three cases where they are superior to at least one of the optimised runs. Moreover, in the forecast phase the non-optimised runs are always superior to the reference runs, but this is not the case in the analysis phase. The reason for this is not completely clear. However, the results do demonstrate that the empirical relationships can improve the forecast skill even when observations are not available.

Figure 4a shows the VOLCAT retrievals in the 13 February 2014 Kelut case study, which we take as representative of the larger magnitude eruptions considered in this paper, at 0530 UTC on 14 February. As also noted by Zidikheri et al. (2017) [3], VOLCAT retrievals covering most of the observed ash cloud were available in the period 13/2330–14/0230 UTC. After this time, however, only partial retrievals, which do not include most of the areas covered by the umbrella cloud (indicated by the red arrow), were available. There is enough data, however, to perform a verification study for the forecast phase even without the umbrella cloud. The mass load forecast obtained from the reference run, comprising the default source term without optimisation, is shown in Figure 4b. We can see that the ensemble mean load in the reference forecast is overestimated by at least an order of magnitude in the region where retrievals are available. The tendency of mass loads in large eruptions to be overestimated (and erroneously distributed) was also noted by Zidikheri et al. (2017) [3], and this continues to be the case here. The forecasted ensemble-mean mass loads for the run employing VOLCAT retrieval-optimised source parameters in real-time are shown in Figure 4c. The agreement between the ensemble mean loads and the VOLCAT retrievals is substantially improved over the reference run, both in terms of magnitudes and spatial distributions. The forecasts employing satellite detections only to optimise the (Figure 4d) and empirical source relationships without optimisation (Figure 4e) also show similar improvements. These results are encouraging because they demonstrate that the empirical relationships between the source term parameters and eruption height, as derived in this study, do encode useful predictive skill since retrieval data at this stage of the eruption was not in any way included in the statistical analysis used to derive these relationships, as alluded to above.

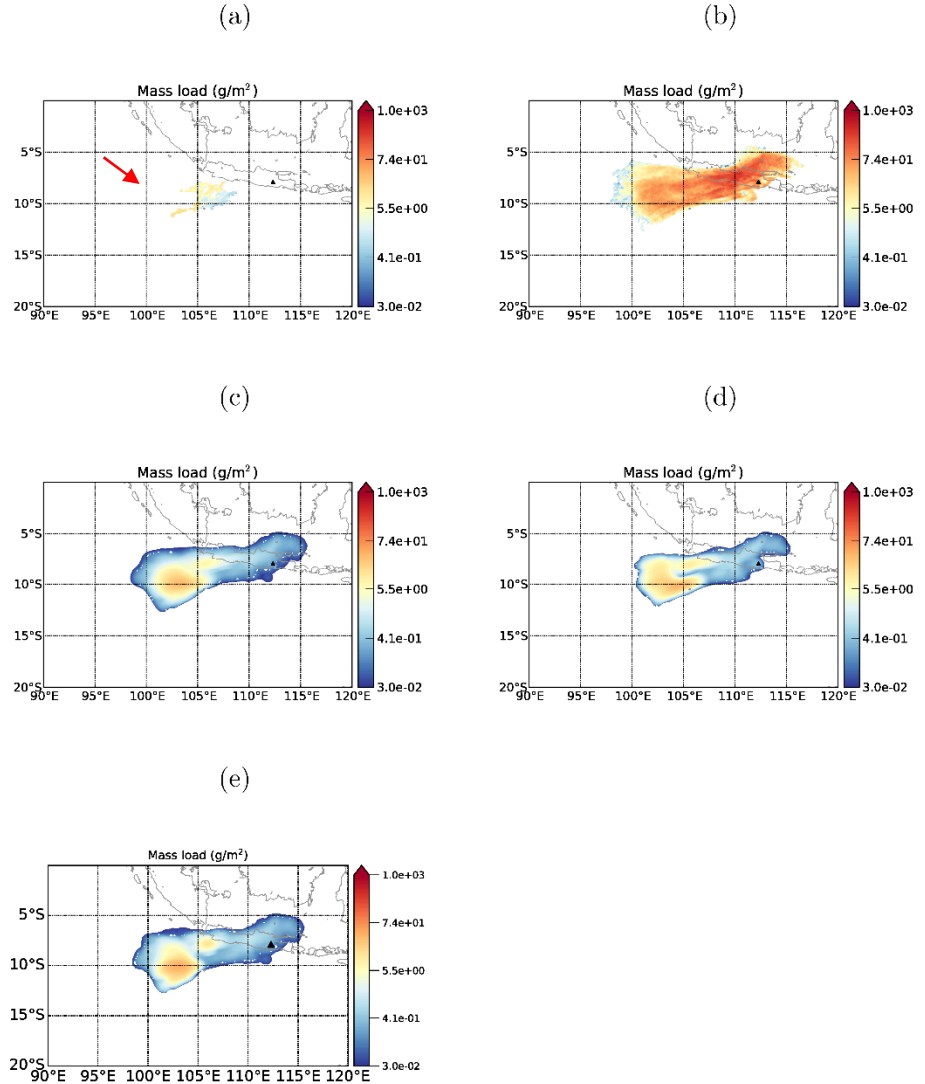

**Figure 4.** Mass Loads at 14/0530 UTC in the February 2014 Kelut case study: (**a**) VOLCAT retrieval, (**b**) reference run ensemble mean, (**c**) retrieval-optimised run ensemble mean, (**d**) detection-optimised run ensemble mean, and (**e**) non-optimised run ensemble mean. The arrow in (**a**) indicates the location of the optically thick ash advected from the umbrella cloud, which was not detected by VOLCAT. The black triangles indicate the volcano location.

VOLCAT retrievals and forecasted mass loads for the May 2019 Agung eruption, taken to represent typical smaller-sized eruptions, at 24/1500 UTC, about 3.5 h after the eruption, are shown in Figure 5. The VOLCAT retrieval at this time (Figure 5a) shows a fairly localised band of ash to the south-west of the volcano with mass loads less than 2 g m$^{-2}$. The ensemble mean mass load from the reference forecast (Figure 5b) predicts the magnitude and general movement of the ash quite well, but is more spread out spatially, with a significant component moving more slowly to the south-west and, therefore, still over land at this time. Both types of optimisations (retrieval-based in (Figure 5c) and detection based in (Figure 5d)) are better able to capture the location of the ash, with a smaller portion being over land even during the forecast phase (about one hour after the latest observation was assimilated—see Table 1). This trend continues until at least 1700 UTC, after which there were no more useful retrievals to verify against. The run employing empirical source relationships but without optimisation (Figure 5e) does not appear to be much better than the reference run in this case. This suggests that in this case most of the improvement in the optimised runs (with respect to the reference run) originate from the optimisation rather than the use of the empirical relationships.

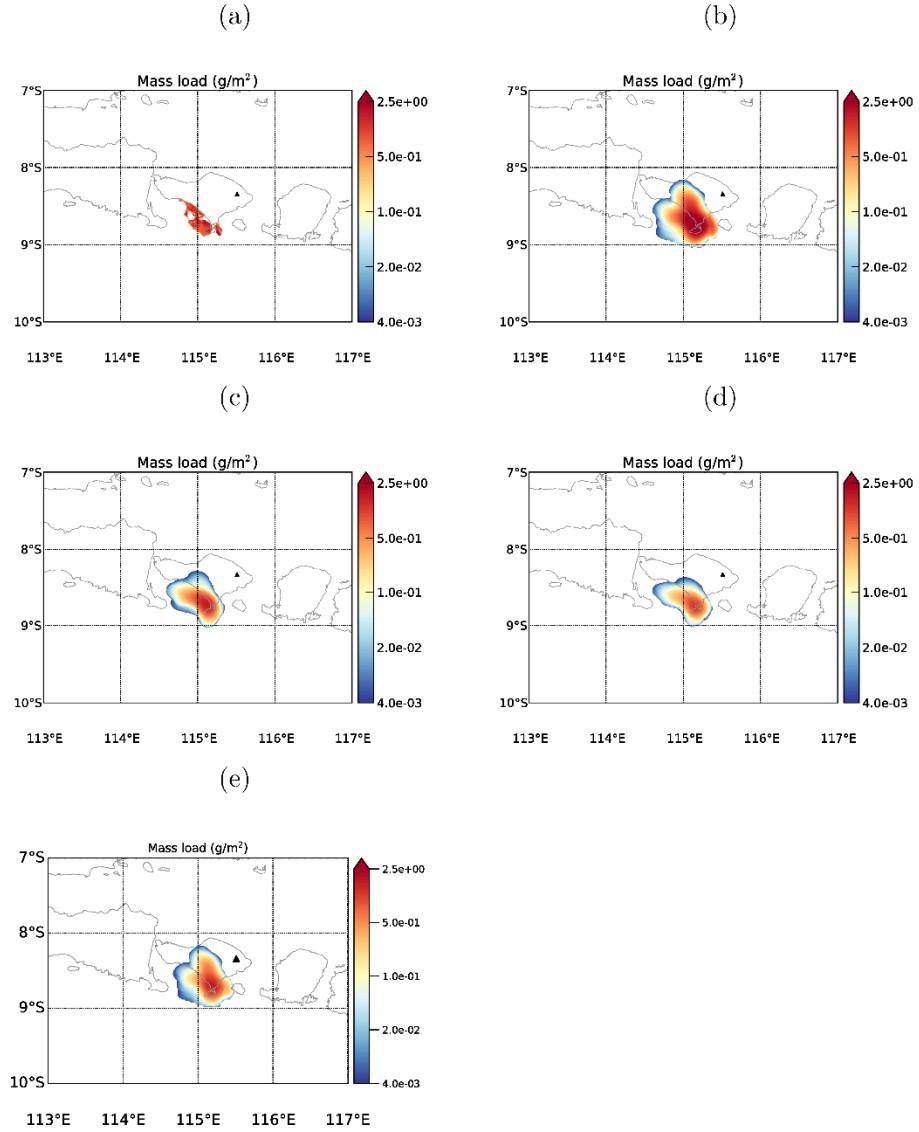

**Figure 5.** Mass Loads at 24/1500 UTC in the May 2019 Agung case study: (**a**) VOLCAT retrieval, (**b**) reference run ensemble mean, (**c**) retrieval-optimised run ensemble mean, (**d**) detection-optimised run ensemble mean, and (**e**) non-optimised run ensemble mean. The black triangles indicate the volcano location.

## 6. Discussion

In this study we have compiled a dataset of NOAA's VOLCAT mass load retrievals from 14 different eruption events that were chosen so that the dataset spanned a wide range of eruption heights. For each event, the inverse modelling procedure of Zidikheri et al. (2018) [6] was used to estimate the range of possible values of eight source parameters, namely the eruption height, $H$, the start-time lag, $\Delta t_0$, the effective umbrella cloud diameter, $D$, the umbrella cloud centre of mass, $\mu_h$, the umbrella cloud depth, $\delta$, the umbrella cloud to eruption column mass ratio, $\beta$, the fine ash fraction, $\epsilon$, and the mean particle radius, $\mu_r$, subject to the satellite retrievals. The average value of each parameter for each case study was then fitted to a power-law relationship with respect to the average eruption height.

The start-time lag, $\Delta t_0$, and the umbrella cloud diameter, $D$, describe the horizontal extent of the ash source. We found that both the start-time lag and the effective umbrella cloud diameter increase with eruption height, which is consistent with the physics of umbrella clouds. Because the start-time lag is simply the time difference between the start of the eruption and the time at which wind-driven

transport becomes dominant, we expect the time lag to increase for large eruptions with strong gravity currents. Similarly, we expect the umbrella cloud diameter, $D$, at this time to be greater for bigger eruptions than for smaller eruptions because of the stronger gravity current.

The three parameters, $\mu_h$, $\delta$, and $\beta$, described the vertical distribution of mass of the source. We found that $\mu_h$ increased with eruption height, which is consistent with plume modelling results. The umbrella cloud centre of mass, $\mu_h$, is expected to be at the level of neutral buoyancy and this will increase with the size of the eruption. Different models predict values ranging from 0.5 $H$ to 0.8 $H$ for this parameter [35], which is broadly consistent with the values of 0.86 $H$, 0.65 $H$, and 0.55 $H$ obtained from the derived power-law relationship in Table 3 for $H$ = 10, 20, and 30 km, respectively. The umbrella cloud 'depth', $\delta$, was found to decrease with increasing eruption height and the umbrella to eruption column mass ratio, $\beta$, was found to increase with eruption height. These results are to an extent a consequence of our modelling choice. We effectively employ two cylinders, one representing the eruption column, and the other representing the umbrella cloud. Large eruptions with large umbrella clouds will have most of the ash mass within the umbrella cloud by the time wind-driven ash transport becomes dominant and, therefore, we expect $\beta$, the mass ratios of the two cylinders, to increase with eruption height. In addition, the mass distribution in the 'umbrella-cloud' cylinder needs to be tightly confined around $\mu_h$, the neutral buoyancy level, to properly model the effect of the umbrella cloud. For smaller eruptions, the umbrella cloud effect is not as strong and, therefore, wind-driven ash transport is the dominant mechanism throughout the length of the eruption column. Therefore, we expect the mass distribution of the source to be more uniform, consistent with larger values of the umbrella-cloud 'depth', $\delta$.

The last two parameters, $\epsilon$ and $\mu_r$, are related to the microphysical properties of the ash. The fine ash fraction, $\epsilon$, describes the fraction of the total emitted ash that remains airborne at time scales at which wind-driven dispersion is dominant. For the most part, this will comprise smaller ash particles as larger particles will be rapidly deposited to the surface. We found that $\epsilon$ decreases with increasing eruption height. The reason for this is not completely clear although such an effect has been suggested by other studies [34]. One explanation is that in large eruptions, particularly in the moist tropics, there are significant amounts of ice embedded within the ash particles, which somehow enhances the fallout of ash particles, therefore reducing the amount of airborne ash. We also found that the mean particle radius, $\mu_r$, increases with eruption height. This might similarly be related to the complex interaction between ash and ice particles. However, testing this hypothesis was outside the scope of this study.

The empirical scaling relationships derived from the compiled dataset can then be used to construct a more computationally efficient data assimilation procedure. Instead of employing thousands of dispersion model simulations to sample multidimensional space for optimal parameter values, the problem is reduced to sampling just the eruption height and the space of possible meteorological fields (provided by the ensemble NWP fields). This is particularly advantageous in an operational setting where fast algorithms are crucial to enable end users maximum time to respond to the issued forecast. The scaling relationships can also be used to provide forecasts of mass loads in situations where satellite retrievals are not available in real time insofar as estimates of eruption height are available. We have shown that this low-dimensional sampling technique, with or without real-time satellite retrievals, leads to significant improvement to the mass load forecast skill, and presumably also to the ash concentration forecast skill.

So far we have not explicitly discussed how temporal variations in eruption strength could be handled within the framework considered in this paper. Most of the eruptions considered here were relatively short-lived and could, therefore, be considered as having constant strength for the duration of the eruption. For long-lived eruptions, such as the November 2015 Rinjani eruptions, we have also assumed that the eruption strength was constant, which was a reasonable first approximation in that case. Clearly, for long-lived eruptions with significant variation in strength, however, this approach might not work very well. A practical solution to this problem is to estimate the eruption height,

$H(t)$, as a function of time from a series of satellite brightness temperatures. These numbers could be input into the model as initial guesses and then optimally rescaled as $\hat{H}(t) = \alpha H(t)$, with the rescaling parameter $\alpha$ taking the place of (the time-invariant) $H$ in the inverse modelling (analysis) phase described in Sections 2 and 5. The other source parameters considered in this paper would of course be estimated as functions of $\hat{H}(t)$ and also become functions of time.

The case studies used to derive the empirical scaling relationships in this study were all chosen to be in the tropics. That means that these relationships implicitly account for typical atmospheric conditions in the tropics and will work best in those regions. To make these relationships applicable outside of the tropics, the coefficients in Table 3 will have to be recomputed using case studies relevant to the region of interest. A natural extension of this study would be to investigate to what extent the predictive skill of the source parameters could be improved by considering not only their dependence on eruption height but also on local atmospheric state variables such as wind, temperature, and humidity. This would make the empirical relationships more generally applicable and possibly more accurate. Another aspect that could be improved in the future is the representation of various source quantities in our model. For example, scaling relationships that relate umbrella cloud diameter to eruption strength and time [36] could be used to improve the crude umbrella cloud representation in this paper. The empirically derived scaling relationships derived in this study have significant errors, particularly for the umbrella cloud depth, mass ratio, and particle size distribution as discussed in Section 4 and readily seen in Figures 1 and 2. This implies in its current form that the search space in our low-dimensional inverse modelling algorithm might be too narrow. In subsequent publications, we shall demonstrate how this might be improved by considering the errors in the optimal source parameter values.

## 7. Conclusions

We have demonstrated how an inverse modelling technique, based on sampling many trial simulations, may be used to infer relationships between various volcanic ash source parameters and eruption height by utilising a dataset comprising satellite mass load retrievals for 14 eruptions. These relationships may then be used to construct more computationally efficient, low-dimensional, data assimilation procedures suitable for operational requirements, with some versions having the added benefit of not requiring real-time mass load retrievals. We have shown that these computationally efficient data assimilation schemes yield mass load forecasts that are significantly improved over current operational settings of the dispersion model.

**Supplementary Materials:** VOLCAT satellite retrieval data used in this study are available at http://doi.org/10.5281/zenodo.3579613. A detailed description of the application of multidimensional inverse modelling to different case studies is available online at http://www.mdpi.com/2073-4433/11/4/342/s1. Figure S1: Mass loads (g m$^{-2}$) in the 13 February 2014 Kelut eruption at 2330 UTC: (a) VOLCAT retrieval, (b) reference run, and (c) experimental run based on inverse modelling. Figure S2: Mass loads (g m$^{-2}$) in the 30 May 2014 Sangeang Api eruption at 1330 UTC: (a) VOLCAT retrieval, (b) reference run, and (c) experimental run based on inverse modelling. Figure S3: Mass loads (g m$^{-2}$) in the second phase of the 30 May 2014 Sangeang Api eruption at 1830 UTC: (a) VOLCAT retrieval, (b) reference run, and (c) experimental run based on inverse modelling. Figure S4. Mass loads (g m$^{-2}$) in the third phase of the 30 May 2014 Sangeang Api eruption at 2130 UTC: (a) VOLCAT retrieval, (b) reference run, and (c) experimental run based on inverse modelling. Figure S5. Mass loads (g m$^{-2}$) at 0630 UTC on 5 November during the November 2015 ongoing Rinjani eruption: (a) VOLCAT retrieval, (b) reference run, and (c) experimental run based on inverse modelling. Figure S6. Mass loads (g m$^{-2}$) at 1630 UTC on 5 November during the November 2015 ongoing Rinjani eruption: (a) VOLCAT retrieval, (b) reference run, and (c) experimental run based on inverse modelling. Figure S7. Mass loads (g m$^{-2}$) in the 31 July 2015 Manam eruption at 0230 UTC: (a) VOLCAT retrieval, (b) reference run, and (c) experimental run based on inverse modelling. Figure S8. Mass loads (g m$^{-2}$) in the 4-6 January 2016 Soputan eruptions at 0030 UTC on 5 January: (a) VOLCAT retrieval, (b) reference run, and (c) experimental run based on inverse modelling. Figure S9. Mass loads (g m$^{-2}$) in the 4–6 January 2016 Soputan eruptions at 0710 UTC on 5 January: (a) VOLCAT retrieval, (b) reference run, and (c) experimental run based on inverse modelling. Figure S10. Mass loads (g m$^{-2}$) in the 1 August 2016 Rinjani eruption at 0600 UTC: (a) VOLCAT retrieval, (b) reference run, and (c) experimental run based on inverse modelling. Figure S11. Mass loads (g m$^{-2}$) in the 20 October 2017 Tinakula eruption at 0130 UTC on 21 October: (a) VOLCAT retrieval, (b)

reference run, and (c) experimental run based on inverse modelling. Figure S12. Mass loads (g m$^{-2}$) in the 11 May 2018 Merapi eruption at 0230 UTC: (a) VOLCAT retrieval, (b) reference run, and (c) experimental run based on inverse modelling. Figure S13. Mass loads (g m$^{-2}$) in the 8 December 2018 Manam eruption at 0530 UTC: (a) VOLCAT retrieval, (b) reference run, and (c) experimental run based on inverse modelling. Figure S14. Mass loads (g m$^{-2}$) in the 24 May 2019 Agung eruption at 1330 UTC: (a) VOLCAT retrieval, (b) reference run, and (c) experimental run based on inverse modelling.

**Author Contributions:** Conceptualization, M.J.Z.; methodology, M.J.Z.; software, M.J.Z; validation, M.J.Z.; formal analysis, M.J.Z.; investigation, M.J.Z.; resources, M.J.Z. and C.L.; data curation, M.J.Z. and C.L.; writing—original draft preparation, M.J.Z.; writing—review and editing, M.J.Z. and C.L.; visualization, M.J.Z.; project administration, C.L. All authors have read and agreed to the published version of the manuscript.

**Funding:** This research received no external funding.

**Acknowledgments:** We would like to thank Richard Dare and Rodney Potts, both from the Bureau of Meteorology, for reviewing the first draft of this paper and providing useful comments. Useful comments were also provided by three anonymous external reviewers and integrated into the final version of this paper.

**Conflicts of Interest:** The authors declare no conflict of interest.

## Appendix A

For completeness, here we present the formulas for estimating the coefficients $a_i$ and $b_i$ in Equation (5), and their standard errors, by weighted linear regression. Equation (8) may be concisely written in vector form as:

$$\widetilde{Y}_i = \chi_i \gamma_i + \widetilde{E}_i, \tag{A1}$$

where

$$\widetilde{Y}_i = \begin{pmatrix} \widetilde{Y}_i^1 \\ \widetilde{Y}_i^2 \\ \widetilde{Y}_i^3 \\ \vdots \\ \widetilde{Y}_i^{N_s} \end{pmatrix}, \chi_i = \begin{pmatrix} w_i^1 & \widetilde{X}_i^1 \\ w_i^2 & \widetilde{X}_i^2 \\ w_i^3 & \widetilde{X}_i^3 \\ \vdots & \vdots \\ w_i^{N_s} & \widetilde{X}_i^{N_s} \end{pmatrix}, \gamma_i = \begin{pmatrix} A_i \\ B_i \end{pmatrix}, \widetilde{E}_i = \begin{pmatrix} \widetilde{E}_i^1 \\ \widetilde{E}_i^2 \\ \widetilde{E}_i^3 \\ \vdots \\ \widetilde{E}_i^{N_s} \end{pmatrix}. \tag{A2}$$

Minimisation of $\widetilde{E}_i^T \widetilde{E}_i$ leads to the well-known linear regression formulae for the optimal value of $\gamma_i$, namely,

$$\hat{\gamma}_i = (\chi_i{}^T \chi_i)^{-1} \chi_i{}^T \widetilde{Y}_i \tag{A3}$$

with covariance matrix:

$$\text{cov}(\hat{\gamma}_i) = \frac{1}{N_s - 2} (\chi_i{}^T \chi_i)^{-1} (\widetilde{E}_i^T \widetilde{E}_i). \tag{A4}$$

Equation (A3) enables the evaluation of the optimal values of the coefficients $A_i$ and $B_i$ in logarithmic space while Equation (A4) enables the evaluation of their standard errors (by computing the square roots of the diagonal elements of the covariance matrix). The corresponding parameter values in normal space, $a_i = 10^{A_i}$ and $b_i = B_i$, together with the respective standard errors, are listed in Table 3 based on 14 eruption case studies.

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
