# Peer review of "Using Satellite Data to Determine Empirical Relationships between Volcanic Ash Source Parameters"

_atmosphere, doi:10.3390/atmos11040342_

Round 1

Reviewer 1 Report

Please see pdf attachement

Reviewer 2 Report

This study provides a useful way of determining default eruption source term parameters which are shown to be superior to current defaults in the region of interest (Darwin VAAC).  This type of work is crucial to the development of quantitative ash forecasts which communicate uncertainty. Thus I recommend it for publication. Please find below a list of corrections, roughly in order of occurrence.

It would be helpful to provide the reader with a glossary of all the terms used in the equations for easy reference. I spent a lot of time combing back through the paper to remind myself what each symbol stood for.

One discussion that is lacking is the connection of plume height to meteorological conditions. Plume height can be significantly affected by meteorological conditions, and so empirically derived scaling relationships which are a function only of plume height might be expected to have relatively large uncertainty when a range of meteorological conditions is represented in the data.  As meteorological conditions such as wind profile and tropopause height are also generally available (as well as forecast) and can influence plume height, a natural extension of this work would be to eventually develop scaling relationships as a function of these variables as well. I would recommend publication by  Tupper, Textor, Herzog, Graf and Richards, “Tall Clouds from small eruptions: the sensitivity of eruption height and fine ash content to tropospheric stability” Nat Hazards (2009) 51:375–401 DOI 10.1007/s11069-009-9433-9 and Bursik (2001) Effect of wind on the rise height of volcanic plumes. Geophys Res Lett 28(18):3621–3624

This connection also suggests that the values presented may not be appropriate for use in all climate conditions or latitudes. Although this does not decrease the usefulness of the paper, there should be room for a discussion of general applicability in the discussion section.  

Line 206-216  I found this sentence confusing “Note, however that E (fine ash fraction) is not determined independently as an additional dimension in parameter space.”  After some thought, I take it to mean that you aren’t performing additional runs with different values of fine ash fraction because one run can effectively represent an infinite number of runs with different fine ash fraction simply by scaling the concentrations according to desired release rate. Then, instead of using the pattern correlation to find the best fine ash fraction among the infinite number of choices, you are using the relationship defined in (4).  I would say that fine ash fraction is still an additional dimension in parameter space which is determined independently, it’s just that you don’t need to perform actual extra runs (scaling the release rate could be seen as a “virtual” run), and you use a different metric to pick the best value.  

Line 232-234. It should be noted that the power law scaling between MER and height is based on buoyant plume theory and the empirical value is very close to what you would get for a simple plume (Sparks “Volcanic Plumes” equation 5.1).  Is there a reason to believe that these other parameters should have a power law relationship with plume height? If not, then perhaps it would be better to say that you simply find that looking for a power law fit is expedient.  

Line 276 – Is this a_i the same as the a_i in equation 9? Also you don’t have a section labeled results. Could you provide the section it is discussed more in?  I do not see it mentioned later.

Line 379-394  In the last part of this paragraph you finally discuss the uncertainty in the Mastin equation. I think this should come earlier. The trial simulation mass loads that are used to compute fine ash fraction still use the Mastin equation (equation 1 with M being described in line 132).  My interpretation would be that your fine ash fraction  is an actual fine ash fraction plus a correction to the Mastin Equation which may be dependent on the type of eruptions that you used for the study. I would recommend publication by  Tupper, Textor, Herzog, Graf and Richards, “Tall Clouds from small eruptions: the sensitivity of eruption height and fine ash content to tropospheric stability” Nat Hazards (2009) 51:375–401 DOI 10.1007/s11069-009-9433-9.  This may offer another explanation for why your fine ash fraction is decreasing with height. It would be interesting to see if there is some connection to the meteorological conditions.

Line 479-481.  I count 5 runs in which the blue is better (1,2,3,12,13).  

Figure 1 and 2.  It would be easier to match the weight with the point if there were points on the red lines.

Figure 1(a) The points for the lowest height values are missing – there are only 12 points.

In Figure 1(c) could you indicate which point is the Kelut eruption?

Figure 2(a)(b)(c) Why do the points with low heights have zero (or almost zero) weighting?

Line3 481-483 Can you clarify this sentence. “The runs not employing optimisation in 481 the analysis phase (but employing empirical source relationships) are generally slightly poorer but 482 perform better in the forecast phase than in the analysis phase.”    I think what you are trying to say is that the yellow runs are better overall (except in three cases (7,8,12 in Figure3(a)) than the reference runs even though they are generally poorer than the optimized runs. When you say they perform better in the forecast phase than in the analysis phase do you mean that their skill score is closer to the red and blue? Or that their skill score is better overall? It makes sense that it would be closer to the red and blue in the forecast phase. I don’t understand why the scores should be higher in the forecast phase although it does look like they might be. If they are, it would be nice if you had an explanation for it.

Figure 4 – I think the date, 13/0530 UTC must be incorrect. In the supplementary material you have the eruption beginning at 16UTC on 13 Feb 2014. 

Figure 4 and 5. Can you include the empirical source parameter relationship runs as well?

Figure 4 (a) colors are very hard to see. Suggest using a different color scale which does not have white in the middle.  Since there is white in the middle, it isn’t clear in (a) where the boundary is. Why is the optically thick umbrella cloud so far away from the location of the volcano? Or is this just an optically thick portion of the cloud? Is it still an umbrella cloud after it has been advected so far downwind?  Should we be seeing higher mass loading in the model runs there?  

Supplemental – again could you change the color scale so there is no white which matches the background in the color scale.

Line 591. I would expect fairly large uncertainty in scaling relationships based solely on plume height. Probably these errors would be even larger if eruptions in a larger latitude range were considered. What about including dependence on meteorological conditions or possibly volcano or eruption type?

Reviewer 3 Report

This is a useful paper which uses inversion techniques for two purposes. The first is to build a dataset of understanding for a range of eruptions and use this to develop parametrizations of volcano behaviour (essentially an attempt to refine the results in Mastin et al and extend them to a wider range of parameters) while the second is a practical method of further improving the descriptions in real time during an eruption. The first aspect in particular could be of wide application and could be used as a "prior model" for other real time inversion methods.

I have some minor detailed comments.

1) I found it quite hard to understand what was going to be done from the abstract and introduction. This all becomes clear later on but it might help to say more early on about what sort of things the authors are considering as "source parameters". This could be a height-time array of emissions (or even a 4-D array to account for the umbrella cloud) or a few simple bulk parameters. E.g. on line 96 "vertical mass distribution" sounds like a grid of values rather than a bulk description given by "parameters of the vertical mass distribution".

2) The approach seems aimed mainly at short eruptions rather than long drawn out eruptions such as Eyjafjallajokull (although the application to Rinjani illustrates what can be done for a long eruption). It might be worth discussing this a little more.

3) Line 43-44: I think the vertical distribution of ash within the rising eruption column is not relevant - one is really interested in the distribution of ash that detrains from the column or is left after the eruption stops. Also it might be worth making a distinction between "fine ash fraction" and "distal fine ash fraction", the later accounting for near source fall out due to ice processes (as discussed) or ash aggregation.

4) Line 124: I think the Ganser formulation requires some shape assumptions on the ash particles. If this is correct the assumptions should be described. If these are very complex just saying that "certain shape assumptions are made as described in ..." would be fine.

5) Generally more needs to be said about the units used. For example on line 132 it is important that H is in km and that the units of M-dot are given (kg/s ?). Units are also needed for table 3.

6) Line 158: this should really be sigma_h << H - mu_h and sigma_h << mu_h - H_0.

7) Line 160: "Particle size distribution" often means "mass distribution". I'm not sure if there is an unambiguous terminology available but could say "particle size number distribution".

8) Lines 192-193: Should there be a particle size cut off to match detection limits of the satellite data? Perhaps this is taken as R = 50 um? I would have thought this was a bit large but it no doubt depends on the satellite instruments and the retrieval/detection method. On line 199, I was slightly surprised that mu_r could affect the detection map (because according to (3) there will always be some ash released at all values of r < R). Probably this is a numerical issue - if g(r) gets very small then in practice there won't be any particles with this value of r. This might explain why mu_r is difficult to determine.

9) Eq (4): epsilon reflects both the fine ash fraction and any error in the Mastin et al eruption rate estimate. That may be what lines 211-213 is saying but that wasn't clear to me.

10) Line 250: Could be more specific and say minimisation of the rms (over j,k) value of E_i^jk.

11) Table 1: I found this hard to understand (e.g. what's the difference between columns 4 and 5?). It all becomes clear later on, but it might help to say columns 5 and 6 are explained/discussed in section 5 below.

12) Line 374: The distribution can be made more or less uniform either by making delta big or beta small. This may explain the large uncertainties in these parameters.

13) Fig 2: In 2(c) the caption and the figure label differ (beta or mu_r). I also thought the red lines were not ideal (although OK) in that there's no reason to join the values up. A series of red dots might be better.

14) Line 435: The reader could check in the references, but it might be useful to say what the authors have in mind (e.g. brightness temperature?, inversion using just detections?).

15) Fig 3: Is it correct to say that the no optimisation results (yellow) are calculated in the same way in the forecast and analysis periods (i.e. the forecast/analysis split is not relevant for the yellow results). If so it might be useful to confirm this.

16) Line 508: Might there be a case for adding the no optimisation case to fig 4 to confirm this? This would just use the empirical relationships without any extra optimisation.

17) Fig 4: I think this should be 14/0530 (13/0530 is before the eruption start).

18) Lines 521-525: Presumably the default D=20km is too big for this case?

19) Eq A1 and A3: I think y_i should be Y_i (or it should be y_i in A2).

Round 2

Reviewer 2 Report

Changes made are acceptable.

Figure 5: Caption needs to be updated  to include description of panel e.

Reviewer 3 Report

I have two comments related to my previous comments 6 and 8 for the authors to consider. However these are relatively unimportant comments and whatever the authors decide is fine.

6) The condition on line 184 is not quite correct. If we have e.g. mu_h = H - sigma_h/1000 then nearly half the umbrella cloud Gaussian would be above H and so would not be included in the integral. Could just add "with mu_h not too close to H_0 or H". Mathematically the correct condition is sigma_h << H - mu_h and sigma_h << mu_h - H_0, but expressing this in words ("mu_h not too close to H_0 or H") might be more intuitive. However this is relativley unimportant - readers would interpret it correctly.

8) Lots of extra material has been added about the size sensitivity of retrievals, which is very good. However it might help the reader a little on lines 192-193 to explain that the choice of R is related to the satellite detection limit (assuming it is).

There are also a few minor editorial (non-scientific) issues to correct. These are not really issues for me as scientific reviewer, and they would hopefully be spotted anyway in proof reading. But, because I've spotted them, I've noted them here.

A) Quite a lot of mathematical symbols are now in bold type for reasons that weren't clear to me. Possibly the intention is to have all the in-line maths in bold and the displayed equations in italic, but this looks strange to me. This is really a style issue for the authors/editors to decide. On line 192, the mu in "mu m" (micrometres) is bold which seems especially strange.

B) Lines 194 and 688: There are (LaTeX?) cross-referencing errors.

C) Line 575: Something's missing in the wording of this sentence, before "(Figure 4(d))".